# Therapeutic Uses of Red Macroalgae

**DOI:** 10.3390/molecules25194411

**Published:** 2020-09-25

**Authors:** Mona M. Ismail, Badriyah S. Alotaibi, Mostafa M. EL-Sheekh

**Affiliations:** 1National Institute of Oceanography and Fisheries, NIOF, Alexandria 21556, Egypt; mona_es5@yahoo.com; 2Pharmaceutical Sciences Department, College of Pharmacy, Princess Nourah bint Abdulrahman University, Riyadh 11671, Saudi Arabia; bsalotaibi@pnu.edu.sa; 3Botany Department, Faculty of Science, Tanta University, Tanta 31527, Egypt

**Keywords:** bioactive compounds, drugs, seaweed, Rhodophyta

## Abstract

Red Seaweed “Rhodophyta” are an important group of macroalgae that include approximately 7000 species. They are a rich source of structurally diverse bioactive constituents, including protein, sulfated polysaccharides, pigments, polyunsaturated fatty acids, vitamins, minerals, and phenolic compounds with nutritional, medical, and industrial importance. Polysaccharides are the main components in the cell wall of red algae and represent about 40–50% of the dry weight, which are extensively utilized in industry and pharmaceutical compounds, due to their thickening and gelling properties. The hydrocolloids galactans carrageenans and agars are the main red seaweed cell wall polysaccharides, which had broad-spectrum therapeutic characters. Generally, the chemical contents of seaweed are different according to the algal species, growth stage, environment, and external conditions, e.g., the temperature of the water, light intensity, nutrient concentrations in the ecosystem. Economically, they can be recommended as a substitute source for natural ingredients that contribute to a broad range of bioactivities like cancer therapy, anti-inflammatory agents, and acetylcholinesterase inhibitory. This review touches on the main points of the pharmaceutical applications of red seaweed, as well as the exploitation of their specific compounds and secondary metabolites with vital roles.

## 1. Introduction

Macroalgae are macroscopic benthic marine algae (seaweed) living in the intertidal zone. They are characterized by autotrophic nutrition and fast-growth; they do not need land for cultivation and their growth rate is faster than terrestrial plants [1]. They are categorized into three different groups Chlorophyta; Phaeophyta, and Rhodophyta, based on their pigment composition and storage food Figure 1.

Red seaweed are the critical source of numerous bioactive compounds, in contrast with the other two groups of green and brown seaweed like polysaccharides “floridean starch and sulfated galactans, such as carrageenans or agars”, minerals, unsaturated fatty acids, amino acids, vitamins, phycobiliproteins, other pigments, phycolectins and mycosporine-like amino acids, which have many biological and industrial applications [2,3].

The protein content in red seaweed varied between 10–50% of the dry weight and being higher than macroalgal groups and some foods [4]. In addition, they contain the essential amino acid, about 25–50% of the total amino acids, is like other protein sources like leguminous plants [5].

Red and green seaweed contain the largest amount of phenolic compounds like flavonoids, phenolics acids, and bromophenols, which had different medical applications, due to the reaction of these components with proteins, e.g., enzymes or cellular receptors. While, phlorotannins, are the major polyphenolic secondary compounds synthesized only in marine brown seaweed [6].

For decades and at present, seaweed is used in food in many countries, as well as in traditional drugs and cosmetics, due to their richness in natural metabolites. The therapeutic trend has begun searching forward for new medications from natural products like algae. Since ancient times, macroalgae are used for treating different diseases. The approximate numbers of biochemicals are more than up 700 from red species. The majority of these contents have shown promising biological abilities, including antimicrobial, antiviral, antitumor, antioxidant, anticoagulant, anti-inflammatory, antidiabetic, antiallergic, and analgesics efficiencies [6,7].

## 2. Antimicrobial Activity

The antimicrobial mechanism of macroalgae depended on the modification of the cell porousness of pathogen species that led to loss of macromolecules and disrupted the function of the membrane and finally destroy the pathogen cells [7,8].

The variation in the biological ability of red seaweed may be related to seasonal variations in algal biological compounds, algal growth phases, and the efficiency of the extraction methods. Carbohydrates, glycosides and phenolic compounds from *Kappaphycus alvarezii* can act as considered as antibacterial agents toward different human pathogenic [9]. *Jania rubens* had antimicrobial activity toward thirty-two multidrug-resistant bacterial isolates which related to its long-chain saturated fatty acids, for example n-hexadecanoic acid or found in ester form such as docosanoic acid 1,2,3-propanetriyl ester and hexanedioic acid, dioctyl ester were responsible for antibacterial ability [8]. Agar and carrageenan oligosaccharides from red seaweed had antimicrobial ability against different human pathogenic microbe [10]. *Gloiopeltis tenax* extracts using supercritical CO_2_ modified with ethanol, primarily composed of fatty acids, phenols sesquiterpenes, ketones and sterols exhibited remarkable antimicrobial activity [11]. Brominated cyclic diterpenes from red alga *Sphaerococcus coronopifolius* has antibacterial activity toward methicillin-resistant bacteria *Staphylococcus aureus* strains [12]. *Laurencia* spp. synthesized halogenated biomolecules with antimicrobial, and cytotoxic potential [13]. Red alga *Plocamium and Chondrococcus* secreted polyhalogenated monoterpenes which exhibit antimicrobial and antitumor [14].

The red seaweed from India Southern Coasts showed higher antifungal ratio “37%” than brown “33.3%” and green seaweed “8.3%” [15]. The Methanolic extract of *Corallina mediterranea*, *Hypnea musciformis* and *Laurencia papillosa* had antifungal activity toward human pathogenic fungi *Aspergillus* and *Candida* [16]. The new diterpene, sphaerodactylomelol fraction 2 extracted from *Sphaerococcus coronopifolius* exhibited a highest antifungal activity toward *Candida albicans* [17]. The antifungal efficacy of methanolic extracts of *Acanthaphora spicifra* had a similar activity of commercial drugs like ciprofloxacin and amphotericin [18].

## 3. Antiviral Activity

Macroalgae have antiviral properties that provide a protective effect against several virus species by obstructing the spread of the human immune deficiency virus (HIV) and other sexually transmitted viruses, such as herpes simplex virus (HSV) and genital warts [19]. Sulfated polysaccharides “SPs” and phenolic compounds from seaweed have better antiviral activity than other bioactive substances via preventing viral adsorption (simultaneous-treatment assay) and replication (post-treatment assay) [20]. The mechanism of macroalgal polysaccharides against viral diseases is focusing on the viral attachment phase via 1) attaching immediately with virions and/or 2) connecting to the respective protein receptors as explained in Figure 2 and Figure 3) immunomodulators that activate Natural killer cells (NK) or prompt immune reactions [21].

Oligopolysaccharides agars and carrageenans from different red seaweed species showed antiviral capacities [3]. The crude polysaccharides from red algae (*Pterocladia capillacea* and *Laurencia obtusa*) from Abo Qir, Alexandria showed antiviral efficiency against hepatitis C (HCV) in vitro [22]. Carrageenans from *Gigartina skottsbergii* may act as models for structuring novel anti- HSV 1 and 2 agents through the mechanism of blocking viral attachment. Also, it could raise the efficiency of NK cells and increase the lymphocytes generation amount [23]. Iota carrageenan from *Solieria chordalis* had antiviral efficiency toward HSV1 [24].

Polyphenol-rich extracts synthesized by Mexican red seaweed *Solieria filiformis* have major virucidal abilities toward the Measles virus and avoid virus dissemination in vitro [25].

## 4. Antioxidant Activity

Antioxidants substances from seaweed have attracted the principal interest in pharmaceutical manufacturing, since these compounds effectively prevent or retard the adverse impacts caused by free radicals. There are two groups of antioxidants (1) the reaction breaking antioxidants and (2) preventive antioxidants [26]. They demonstrate the main roles in the prohibition of many maladies such as a tumor, inflammatory disorders, coronary heart, neurological degeneration, and aging [27].

There are many specialists occupied with discovering natural antioxidants posing security and adequacy, which can be replaced the manufactured ones butylated hydroxyl anisole (BHA), butylated hydroxyl toluene (BHT), α-tocopherol and propyl gallate which led to destroying the liver and causing cancer [28]. Ina and Kamei [29] classified the algal antioxidants into two groups; (1) water-soluble antioxidants vitamins, phycobiliproteins and polyphenols, and (2) fat-soluble antioxidants carotenoids and vitamin E (α-tocopherol).

Carrageenans, porphyrans, and agars have strong antioxidant and immunostimulatory [30]. The antioxidant activity of polysaccharides relies upon the level of sulfate, molecular weight, sort, and branching of the main sugar. Hence, the antioxidant efficacy of low molecular weight SPs is better than those of high molecular weight SPs whereas, the smallest polysaccharides may integrate into the cells more effortlessly and give protons higher adequately contrasted with larger polysaccharides [31]. Benattouche et al. [32] recommended the SPs from *Corallina officinalis* for use in natural antioxidants in food industry applications.

Phycobiliproteins are hydrophilic groups of pigment-proteins from red macroalgae, have antioxidant ability in vitro and in vivo toward free radicals and selenium [33]. R-phycoerythrin from *Mastocarpus stellatus* has antioxidant ability [34]. R-phycoerythrin from *Palmaria palmata* and *Polysiphonia urceolata* identified as antioxidant agents [35]. Mycosporine from red seaweed had antioxidant activities [3].

Phenol compounds play a vital role against external stress. These compounds vary from simple acids to more complex polyphenolic compounds. These compounds are great antioxidants and act as chelators of metals and free radicals, inhibiting lipid, and scavengers of Reactive Oxygen Species (ROS) like H_2_O_2_; OH^•^; O_2_^•−^ & 1O_2_ [6]. The phenolic and flavonoid contents of red seaweed from the Alexandria coast are significantly correlated to their DPPH antioxidant ability [2,36]. The antioxidant potency of macroalgal phenolic compounds counteracts the deleterious impacts of the advanced glycation end products (AGEs), which are the toxic effects associated with hyperglycemia (Figure 3) [6]. The red alga *Symphyocladia latiuscula* bromophenols had antioxidant efficiency [37]. The antioxidant capacity of *Polysiphonia stricta* relates to brominated units and its degrees [38].

Despite the low macroalgal lipid content but it is characterized by a high content of polyunsaturated fatty acid PUFA, especially the arachidonic, eicosapentaenoic, α-linoleic, and octadecatetraenoic acids which had the prohibition of cardiovascular infections, diabetes, antimicrobial antiviral and antioxidant abilities [39]. The antioxidant activities of the extracted fatty acids from *Pterocladia capillacea* and *Osmundaria obtusiloba* were more than 50% Ascorbic Acid Activity (AsA) [40]. Bioactive peptides from red seaweed *Polysiphonia urceolata* and *Palmaria palmata* are known as antioxidant agents [35]. The protective role of lipophilic constituents of macroalgae toward lipid peroxidation is related to low polarity and high solubility [26].

Also, enzymes from red algae *Palmaria palmata*, such as proteases and carbohydrases showed antioxidant efficiency [41].

## 5. Anticancer and Antiproliferative Activities

Cancer is a noteworthy medical issue worldwide and, until now, lacks fully effective medicines. Macroalgae have cancer-fighting agents that may demonstrate valuable in relieving tumors and other cancer conditions like breast, colon cancer, and leukemia [19,42]. In the 1960s, the first anticancer medicine from marine algae (algasol T331) has been detected in Italy. Recently, new anticancer medicine was be produced from red seaweed *Gracilaria foliifera* and green seaweed *Cladophoropsis* sp. [43].

The antioxidant efficiency of the polysaccharide fractions itself might play a role in their antitumor action. Polysaccharides from *J. rubens* were a potential candidate for anticancer treatment [42]. Carrageenan oligosaccharides from different red species showed anticarcinogenic action with less cytotoxicity and synergistic impacts with usual medicines, as well as enhancing the immunocompetence destroyed cells by these mediciness [44]. Luo et al. [45] recommended that the λ-carrageenan injection that could repress malignant development in B16-F10 and 4T1 bearing mice and upgrade tumor resistant reactions by rising the tumor-infiltrating M1 macrophages number dendritic cell antigen “DCs” and extra stimulated CD4 + CD8 + T lymphocytes in the spleen. Agar extracted from cold-water extraction of red algae *Gracilaria* species exhibits antitumor efficiency with antioxidant activity [37]. Porphyrans are an anionic polysaccharide, isolated from *Porphyra* sp. shows anticancer activity [46]. Coura et al. [47] reported the biological activities of agars oligosaccharides anti-tumoral and immunomodulatory.

Phycobiliproteins from red macroalgal species have an important role as anticancer agents, due to their high efficiency and low toxicity. They can enhance the activity of conventional anticancer medicines, decreasing their side impacts, and act as photosensitizers for the infected cells treatment [48]. Especially, phycocyanin (PC) has the antitumor impact that can obstruct the cancer cells multiplication and kill the infected cell, thereby PC can serve as a promising anticancer agent [49]. PC exhibited many anticancer mechanisms, such adjusting the mitochondrial membrane potential (MMP), which invigorates the produce of cytochrome c and elevate the ROS formation, at last lead to cancer cell apoptosis. Also, it can repress the Cyclooxygenase-2 (COX-2) and prostaglandin E2 (PGE2) definition and down-regulate the MMP-9 explanation via mitogen-activated protein kinase (MAPK) signaling pathway. Moreover, they are a toxin on tumor cells and are non-toxic to ordinary cells [50]. The anti-proliferation activity of PC is mediated by BCR-ABL signaling and suppression of the downstream PI3K/Akt “Phosphatidylinositol-4,5-bisphosphate 3-kinase/ protein kinase B” pathway. Also, (MAPK), (PI3K/Akt/mTOR the mammalia target of rapamycin), and Nuclear Factor (NF-κB) pathways involved phycocyanin-induced cell death [49].

Macroalgal phenolic compounds play an important role as tumor fighter through inhibition of the tumor cells via a xenobiotic processing, which changes the carcinogen activation, or through disturbing the cellular division during mitosis at the telophase stage. They also decrease the amount of cellular protein and mitotic index, while some flavonoids can alter hormone production and inhibit aromatase to restrain the malignancy cells progress [51]. Terpernoids, isolated from red seaweed, have demonstrated a high antitumor ability [37]. The water extracts of *Laurencia obusta* indicated significant antiproliferative ability toward both HCT15, A549 and MCF7 due to their polyphenols contents [52]. The bis (2,3-dibromo-4,5-dihydroxybenzyl) ether from most red seaweed had apoptotic ability toward K562 human myelogenous leukemia cells [53]. The organic extract from *Rhodomela confervoides* contain 3-bromo-4,5-dihydroxy-benzaldehyde bromophenols and 3-bromo-4,5-dihydroxy benzoic acid methyl ester with a maximum activity toward KB, A549 cancer cell lines and Bel-7402 (Human papillomavirus-related endocervical adenocarcinoma) [38]. The diterpene, sphaerodactylomelol fraction 1 from *Sphaerococcus coronopifolius* inhibited the cell proliferation [15].

## 6. Anti-Inflammatory Activity

Inflammation is defined as a defense reaction in a broad assortment of physiological and pathological procedures, caused by injury, harm, and contagion, or it may be due to the release of chemical substances from migrating cells [54].

Macroalgal polysaccharides exhibited anti-inflammatory potency with no toxic effects on human health. Orally administration of polysaccharide fractions from *Gracilaria verrucosa* to mice and intraperitoneally increased the anti-inflammatory and antioxidant capacities and stimulating phagocytosis [55]. A sulfated polysaccharide fraction from *Gracilaria cornea* shows anti-inflammatory activity via inhibition of histamine, neutrophil migration, and vascular permeability [47]. Porphyrans extracted from *Porphyra* sp. shows anti-inflammatory activity in humans by the suppression of nitric oxide (NO) production and hindering NF-B activation in the mouse macrophages of RAW264.7 cells [46]. Carrageenans from red species act as anti-inflammatory inducing agents in experimental animals [56].

Phycobiliproteins are commercially used for their therapeutic value as anti-inflammatory agents [57]. Especially R-phycoerythrin from most red seaweed such as *Gelidium pusillum*, *Chondrus crispus*, and *Gracilaria verrucosa* exhibited anti-inflammatory capacity [58]. Phycocyanin exhibited an anti-inflammatory activity with antioxidative ability [37].

PUFA from red seaweed had anti-thrombotic and anti-inflammatory properties [59]. Van Ginneken et al. [60] recommend that the n-6/n-3 ratio between fatty acids should < 10 in the diet, which preventing cardiovascular, inflammatory, and nervous system disorders.

## 7. Analgesic and Antipyretic Activities

Algal terpenes, peptides, and sulfated polysaccharides act as painkiller agent [61]. The red alga *Dichotomaria obtusata* aqueous extract inhibited the production of endogenous mediators in response to acetic acid, due to its metabolites, such as polysaccharides and phenols [62]. The analgesic activity of red algae *Vidalia obtusaloba* and *Ceratodictyon spongiosum* was related to bromophenolic and peptide metabolites [63].

Antipyretics reduce the elevated body temperature. Macroalgal bioactive compounds act as novel and safe for antipyretic agents, e.g., flavonoids like baicalin, which indicated antipyretic impact by suppressing Tumor Necrosis Factor (TNF-α) and hindrance of arachidonic acid peroxidation that decrease the prostaglandin ratio and reduce fever and pains [64]. The possible antipyretic mechanism of macroalgae may be due to the inhibition of prostaglandin, such as paracetamol by blocking the cyclooxygenase enzyme activity and/or inhibition of any of mediators of pyrexia [65]. The antipyretic capacity of *Hypnea musciformis* and *Gracilaria dura* methanolic extract was dose-dependent on albino mice and caused a decrease in body temperature up to 4 h following its administration, due to the inhibition of prostaglandin synthesis, compared with standard paracetamol [66].

## 8. Anticoagulant and Antithrombotic Activities

Anticoagulants are defined as a substance that treats or stops blood clots and minimizes the risk of stroke, cardiac disappointment, and obstruction within blood vessels. The anticoagulant capacity of seaweed may be attributed to its polysaccharides composition, molecular weight, sulfate content, and position, e.g., uronic acids, carrying a negative charge, which gives it the capacity to bind calcium ions, and therefore, prevents the formation of a clot [67]. The algal anticoagulation mechanism may be due to their direct impact on thrombin and enhancing of antithrombin III.; Moreover, algal polysaccharides delayed activated partial thromboplastin time (APTT), proposing the obstacle of intrinsic factors, extended essential pathway-dependent coagulating times, and decreased platelet aggregation.

Commonly, galactans from red seaweed could be alternative sources of new anticoagulant agents [68]. Sulfated galactans from *Grateloupia indica* had anticoagulant efficiency as heparin [69]. Carrageenans from red species contains –O-SO_3_H group, which plays a vital role in blood clotting inhibition by inhibiting platelet aggregation [70]. Carrageenans had about one-fifteenth of the heparin action. The λ-carrageenan exhibited superior anticoagulant ability than κ-carrageenan relating to its higher sulfate content [71]. Depolymerization of agars extracted from *Porphyra yezoensis* and *Gracilaria birdiae* by ultrasound assisted increased the anticoagulant activity [47].

## 9. Antidiabetic Activity

Diabetes mellitus is a metabolic disorder resulting from an imperfection in insulin secretion and/or insulin activity. It is an almost coarse serious metabolic disease in the world.

Seaweed species have a unique antidiabetic way in search of natural alpha-glucosidase inhibitors that reduce the absorption of glucose from the gut itself [6].

However, seaweed species have α-amylase and α-glucosidase inhibitory actions [72]. The sulfated galactans extracted from *Gracilaria opuntia* was responsible for the antidiabetic properties through the deactivation of α-glucosidase, a-amylase, and dipeptidyl peptidase-4 [73]. *Laurencia dendroidea* could be a natural source for the production of antidiabetic agents [74]. Chen et al. [75] demonstrated glucosidase inhibitory action of agar polysaccharides, which increased the acid hydrolysis process.

Mittal et al. [57] demonstrated the antidiabetic efficiency of R-PE and R-PC from *Chondrus crispus*, *Gelidium pusillum*, *Heterosiphonia japonica*, and *Palmaria palmata*. The organic extracts of red algae *Gracilaria changii*, *G.; lemaneiformis*, *Gelidium amansii*, *Osmundea pinnatifida*, containing phenolic compounds, showed glucose uptake regulation [76]. There are other active compounds extracted from Rhodophyta spp. (*Polysiphonia urceolata*, *Rhodomela confervoides* and *Symphyocladia latiuscula*) exhibited antidiabetic properties like bromophenols, benzene acetamide, 2-piperidione, and *n*-hexadecanoic acid [77].

## 10. Anti-Obesity Activity

Obesity is excessive fat accumulation, and is known to increase the risk of many dangerous diseases like type II diabetes, hypertension, hyperlipidemia, and cardiovascular diseases. Obesity is gaining increased attention because of the high expense and dangerous symptoms of anti-obesity drugs. Rhodophyta species were shown to have anti-obesity properties [78].

The ethanolic extract of *Grateloupia elliptica* (60%) reduced the accumulation of the lipid in 3T3-L1 cells and inhibited the adipogenic proteins expression. In addition to a significant decreasing in body weight of C57 BL/6J male mice, as well as reducing white adipose tissue (WAT) weight, e.g., fatty liver, leptin, total cholesterol, and serum triglycerides contents in vivo without cytotoxic effect [79]. Forty % of *Plocamium telfairiae* ethanolic extract showed anti-obesity ability via reducing the fat accumulation and suppressed the expression of major adipogenesis factors, like peroxisome proliferator-activated receptor-γ (PPAR-γ),CCAAT/enhancer-binding protein (C/EBP)-α, sterol regulatory element-binding protein 1 (SREBP-1), and phosphorylated ACC (pACC) in 3T3-L1 cells [80]. Seo et al. [81] demonstrated the antidiabetic activity of extract from *Gelidium amansii* via reduction the accumulation of lipid in 3T3-L1 adipocyte cell line.

Generally, marine algal polysaccharides are considered to be dietary fibers so they are not digested by humans [82]; hence SPs can hinder adipogenesis through the mitogen activated protein kinase (MAPK) in 3T3-L1 pre-adipocytes [83].

## 11. Antihypertensive Activity

Seaweed exhibits significant anti-hypertensive activities [84]. Macroalgae consumption led to decreased blood pressure, which might be linked to the hypotensive effects of the dietary fiber and their rich nitrate content [19]. The antihypertensive effects of macroalgal peptides maintained a healthy heart by stimulating circulation in the blood vessels, and avoiding deadly conditions, such as heart breakdown, atherosclerosis, and peripheral vascular disease [85].

The secondary metabolites of seaweed act as hypoglycemic agents, reduce blood pressure and regulate cholesterol levels, inhibition of hepatic cholesterol biosynthesis, also for hyperplasia prevention, gastrointestinal, regenerative Nori-peptides from *Porphyra yezoensis* have the important antihypertensive ability in hypertensive patients, as well as spontaneously hypertensive rats [86].

## 12. Acetylcholinesterase Inhibitory ‘’Alzheimer’s Disease”

Alzheimer’s disease (AD) is a progressive and degenerative problem in brain regions, chiefly campus, and neocortex responsible for mental functions that reduced neurotransmitter acetylcholine (ACh). AD can prompt amnesia, abnormalities, and cognitive disturbances [87]. In the cholinergic theory, serious damage of cholinergic neurotransmitter AChE in the central nervous system (CNS) gives AD indications [87]. The principle therapeutic strategy against AD is acetylcholinesterase hindrance.

There is very few research reporting on the AChE inhibitory (AChEI) impact of seaweed. The AChEI ability of plant extracts is classified into potent inhibitors (> 50% inhibition), moderate inhibitors (30–50% inhibition), and weak inhibitors (<30% inhibition) [88]. As clarified in Figure 4, *Ochtodes secundiramea* extracts exhibited moderate potency (48.59 ± 0.8%), while the red algae *Hypnea musciformis* (7.21%) and *Pterocladia capillacea* (5.38%) extracts had a weak action. The AChEI abilities of these algae are related to solely compose of halogenated monoterpenes [89].

The alcoholic extract of *Gracilaria corticata* (IC50 9.5) and *G.; salicornia* (IC50 8.7 mg/mL) extracts showed moderate AChEI efficiency [90]. The sulfated polysaccharides from *Gelidium pristoides* exhibited inhibitory potency on acetylcholinesterase that related to their antioxidant and neuroprotective potentials [91].

Some red seaweed species synthesize homotaurine, aminosulfonate compounds, which could be a promising medicine for Alzheimer’s disease prevention [92].

## 13. Macroalgae for Skincare

Macroalgae metabolites reduced the appearance of redness and blemishes, the appearance of sun damage, brightening, re-mineralizing, hydrating and firming skin [93]. They have a reaction mechanism toward the hazardous ultraviolet ‘UV-A and -B’ impacts via delivering phenolic compounds, mycosporine, amino acid, and carotenoids, which act as UV-absorbing [94]. The extracts of red seaweed *Asparagopsis armata*, *Gelidium corneum*, *Corallina officinalis* had skin softness, whitening/lightening, and elasticity restoring anti-aging properties so they can be used as skincare products including creams, oil, soap, mask, or lotion [95]. Agarose from *Gracilaria cornea* and *G.; lemaneiformis* are used for skin whitening, due to its anti-melanogenic activity by inhibiting melanin synthesis [96,97]. Fatty acid-like palmitic acid and its derivated ascorbyl palmitate from seaweed are used in cosmetics as emulsifiers and antioxidant agents for anti-wrinkle and anti-aging characteristics [98]. Amino acid extracted from *Asparagopsis armata* is inserted in some anti-aging lotions [37]. Mycosporine from different Rhodophyta species act as photoprotective substances for skin care products with antioxidant properties [3].

## 14. Conclusions

This review aims to provide an overview of some medical and cosmetic importance of red macroalgae. They represent renewable biomass and are a potentially fruitful source of vital bioactive compounds that ecofriendly and safe. These bioactive compounds are already used mainly in food and medicine. For example, sulfated polysaccharides from red algae are commercially applied as a useful ingredient as a thickening agent, stabilizer, emulsifier, texture modifier, and dietary fiber in food and pharmaceutical industries. There are some species from Rhodophyta whose feasibility medical potential is higher, like *Gracilaria* spp., *Pterocladia* spp. *Jania* spp. and *Corallina* spp. Due to the economic importance of seaweed, more studies should be undertaken, focusing on improving seaweed production on a large scale by adjusting the culture conditions. Cultivation of economic species in seaweed aquaculture may be the future road for the sustainability of seaweed and controlling the production of active compounds. Optimization extraction methods, purification, and fractionation of bioactive compounds led to the production of more active and safe compounds. Therefore, more clinical studies should be carried out on a large scale for economic production.

## Figures and Tables

**Figure 1 molecules-25-04411-f001:**
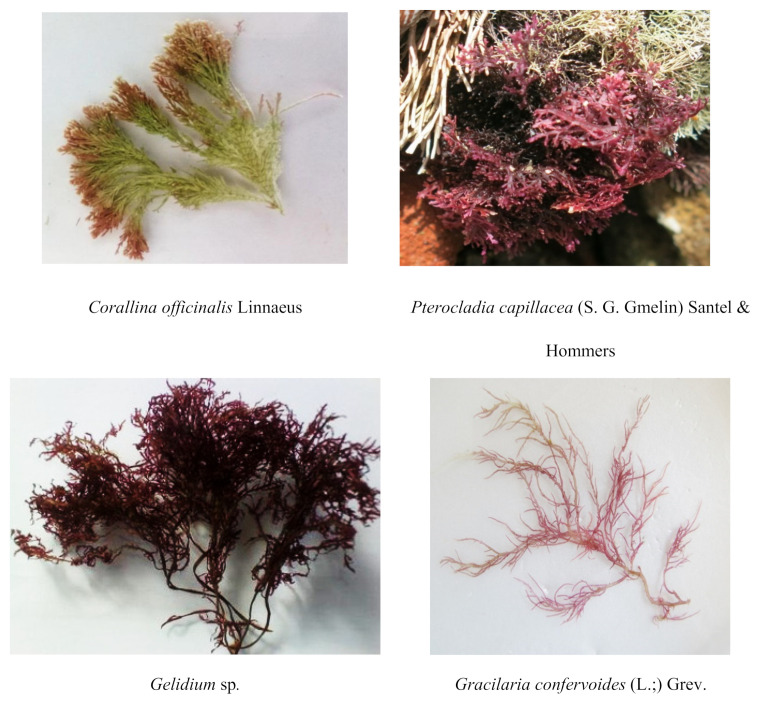
Photo of common red seaweed species with potential from Egypt coasts.

**Figure 2 molecules-25-04411-f002:**
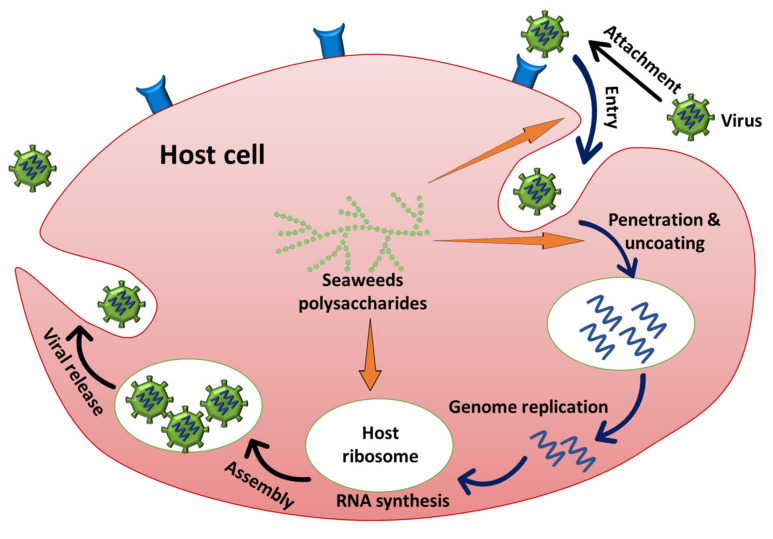
The viral infection stages and the antiviral potency of macroalgal polysaccharides, modified after [21].

**Figure 3 molecules-25-04411-f003:**
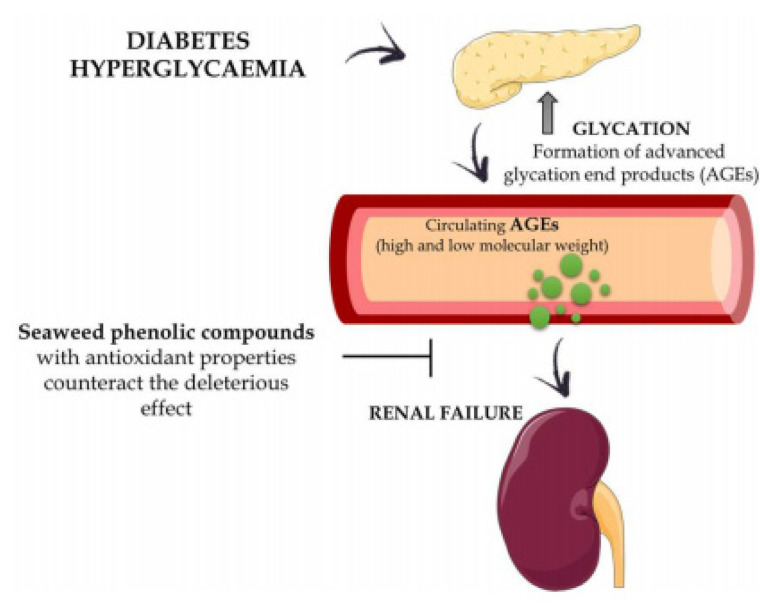
Graphical illustrating the antioxidant activities of macroalgal phenolic compounds toward the harmful impacts related to the hyperglycemia [6].

**Figure 4 molecules-25-04411-f004:**
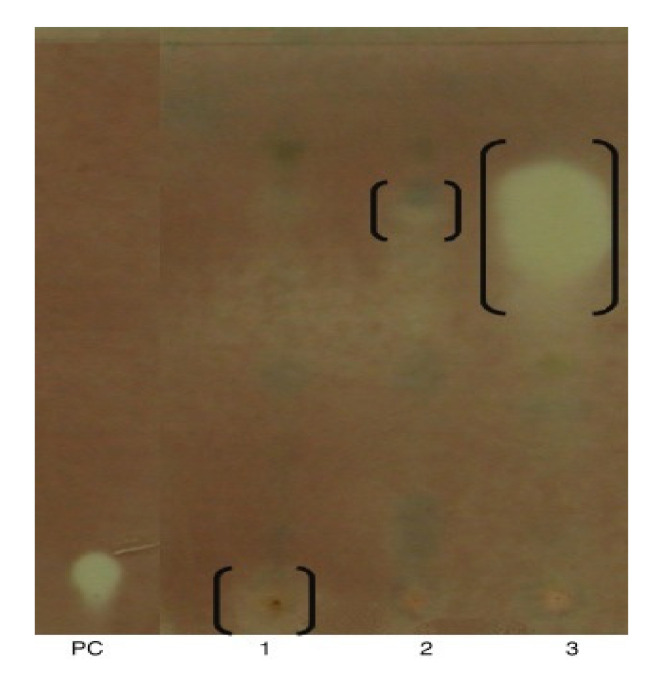
TLC qualitative AChEI assay. PC: +ve control, physostigmine (0.03 μg). DCM/MeOH extracts (100 μg) of (1) *Hypnea musciformis*, (2) *Pterocladia capillacea*, and (3) *Ochtodes secundiramea*. TLC elution system: hexane:ethyl acetate:methanol (2:7:1 *v*/*v*/*v*) [89].

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
