# Peer review of "Therapeutic Uses of Red Macroalgae"

_molecules, 2020, doi:10.3390/molecules25194411_

Round 1

Reviewer 1 Report

Therapeutic uses of red macroalgae are well reviewed in this manuscript. Some minor revisions are needed.

  1. line 42: " Red and green seaweed contain the largest proportion of phenolic compounds " As we know, brown seaweed contain rich phenolic compounds, even more than rad and green seaweed. Please check mmore reference and revise this sentence.
  2. line 70: " The red seaweed showed higher antifungal ability than brown "33.3%" and green seaweed  "8.3%" " This sentence is hard to be understand. Why the red seaweed showed higher antifungal ability? What's the meaning of different percentages?
  3. line 129-130: Please give more informationof the relationship of  antioxidant capacity relating to units and degrees of brominated.

  4. line 318: " Agarose from Gracilaria cornea and G. lemaneiformis are used for skin whitening". Why? please give mor information.

Author Response

Reviewer 1

  1. line 42: " Red and green seaweed contain the largest proportion of phenolic compounds " As we know, brown seaweed contain rich phenolic compounds, even more than rad and green seaweed. Please check mmore reference and revise this sentence.
  • Brown algae contain a biggest portion of polyphenolic compound e.g. phlorotannins"a group of complex polymers of phloroglucinol (1,3,5-trihydroxybenzene)" which are found in neither green nor red seaweeds. (Corona et al. 2017). Phlorotannins are the major polyphenolic class found only in the marine brown seaweeds (Cotas et al., 2020)
  • The largest quantity of phenolic compounds were detected in  green and red seaweeds are of bromophenols, flavonoids, phenolics acids, phenolic terpenoids, and mycosporine-like amino acids (Wells et al., 2017; Corona et al. 2017; Heo et al., 2005). Bromophenols were first isolated from the red alga Neorhodomela larix(Rhodomela larix) (Katsui et al., 1967). There are different bromophenols identified in Rhodophyceae species also detect in brown seaweeds, like bis-(2,3-dibromo-4,5-dihydroxy-phenyl)-methane [256].   

 Corona, G.; Coman, M.M.; Guo, Y.; Hotchkiss, S.; Gill, C.; Yaqoob, P.; Spencer, J.P.E.; Rowland, I. Effect of simulated gastrointestinal digestion and fermentation on polyphenolic content and bioactivity of brown seaweed phlorotannin-rich extracts. Mol. Nutr. Food Res. 2017, 61.

Cotas J., Leandro A., Monteiro P., Pacheco D., Figueirinha  A., Gonçalves AMM, da Silva GJ and Pereira L (2020). Seaweed Phenolics: From Extraction to Applications. Mar. Drugs. 18(8): 384.

Heo, S.J.; Park, E.J.; Lee, K.W.; Jeon, Y.J. Antioxidant activities of enzymatic extracts from brown seaweeds. Bioresour. Technol. 200596, 1613–1623.

Katsui, N.; Suzuki, Y.; Kitamura, S.; Irie, T. 5,6-dibromoprotocatechualdehyde and 2,3-dibromo-4,5-dihydroxybenzyl methyl ether: New dibromophenols from Rhodomela larixTetrahedron 196723, 1185–1188. 

Wells, M.L.; Potin, P.; Craigie, J.S.; Raven, J.A.; Merchant, S.S.; Helliwell, K.E.; Smith, A.G.; Camire, M.E.; Brawley, S.H. Algae as nutritional and functional food sources: Revisiting our understanding. J. Appl. Phycol.201729, 949–982.

  1. line 70: " The red seaweed showed higher antifungal ability than brown "33.3%" and green seaweed  "8.3%" " This sentence is hard to be understand. Why the red seaweed showed higher antifungal ability? Rephrased
  2. What's the meaning of different percentages?
  • This percentage explains the inhibition ratio of antifungal activity of different seaweed.

  1. line 129-130: Please give more informationof the relationship of  antioxidant capacity relating to units and degrees of brominated.
  • Brominated unit and degrees

    line 318: " Agarose from Gracilaria cornea and G. lemaneiformis are used for skin whitening". Why? please give mor information

         Due to its anti-melanogenic activity via inhibit melanin synthesis [94, 95].

Reviewer 2 Report

General remarks:

A worthy effort on an important area of research, providing consistent information regarding medical and cosmetic applications of Rhodophytes.

The references cited are recent and balanced, although there is so much information regarding the subject that many other references exist and could have been cited.

Some English reviewing is required because some sentences are difficult to understand.

Also, most sections seem just a compilation of sentences, copied from different papers, with no critical analysis from the authors. I believe, thus, that a major effort must be made to review the paper, rephrasing the paragraphs, and making a more mature/critical judgment of the information searched.

Also, in each section, a few species are mentioned as examples of the bioactivities referred, which is correct. But the way the text is written, it seems that only those species have such bioactivities. I suggest some rephrasing in each section, addressing this issue.

Thus, although I acknowledge the effort and the importance of the matter, I recommend a major revision of the paper.

Specific remarks are made below to improve the manuscript.

Specific remarks.

Abstract

Remove the “” from Rhodophyta and from carrageenans and agars.

Introduction

Line 33. I disagree that red seaweed are the “best critical source” for the other two groups also present very important compounds. I suggest you remove the word “best”.

Line 36. I suggest rephrasing “pigments, phycobiliproteins” to “phycobiliproteins and other pigments”.

Line 40. Your examples are rather strange. I suggest you rephrase into “animal or plant origin such as meat, eggs or leguminous”.

Line 42. I disagree with the statement “Red and green seaweed contain the largest proportion of phenolic compound”. And that is not what reference [6] says: “The largest proportion of phenolic compounds contained in green and red algae are bromophenols, phenolic acids, and flavonoids.” Actually, brown usually have the highest content in phenolics. I suggest rephrasing.

Line 52 – You should make a general statement about the ability of the red seaweeds to control/kill microbes. The way you wrote it, it seems that only a few species have an antimicrobial activity which is not true – many others also have it and are not mentioned in the text.

Lines 53-55 – please rephrase into a more correct English form, because I can’t understand what you mean.

Line 57 – correct Jaina to Jania

Line 58 – Who are “they”?

Line 65 – Actually new molecules with antimicrobial and cytotoxic activities were identified by Rodrigues et al 2015 for Sphaerococcus coronopifolius (Mar. Drugs 201513(2), 713-726; https://doi.org/10.3390/md13020713). This is an interesting reference to cite both in this section and in the anticancer section.

Line 70 – the sentence makes no sense with “33.3%” and “8.3”. What do you mean?

Line 71 – because it is the first time you cite Cy. crinita you must write the full scientific name (the two epithets).

Antiviral activity

Line 89 – remove “”; both agars and carrageenans should be plurals because there are different molecules of each produced by red seaweeds.

Line 91 – correct obtuse to obtusa

Line 96 – remove “” from the species name Solieria filiformis

Line 97 – Bromoform, produced by Asparagopsis sp. also has interesting antiviral properties that you may refer (besides reducing methane production).

Antioxidant activity

Line 105-109 – remove all “”

Line 110 – again, Carrageenan, porphyrin, and agar should be plurals.

Line 116-117 – this sentence doesn’t add any new information from the sentence in line 110. I suggest you remove it.

Line 118 – What do you mean by “one of the best groups of pigment proteins from red macroalgae”? As far as I know, red seaweeds present only this group of protein pigments. I suggest rephrasing.

Line 124 – If you decide to keep the sentence in lines 116-117 “Reactive Oxygen Species“ should go up there.

Line 126 – DPPH, as ABTS, TPC, or FRAP, are methods to assess antioxidant ability, so antioxidant compounds are measured using such methods. I don’t really understand what you mean by this sentence.

Line 132 – remove “” and substitute by (). Rephrase the sentence into a more correct English.

Line 135 – substitute P. capillacea by full scientific name.

Line 136 – 50% of what??

Line 139 – correct Parmaria palmate to Palmaria palmata

Anticancer and antiproliferative activities

Line 146 - – remove “” from both species names

Line 163 – the sentence “They are also used as fluorescent indicators in immunological and clinical diagnostics” makes no sense in the context of this section. I suggest deleting it.

Line 187 – the sentence “Mycosporine from red seaweed had antioxidant activities [3]” makes sense in the antioxidant section, not here.

Anti-inflammatory activity

Line 193 – link references 53 and 56, because the anti-inflammatory effect of G. verrucosa (that currently is known as Gracilariopsis longissima) may be due to the presence of R-phycoerythrin.

Analgesic and antipyretic activities

Line 209 – rephrase the first sentence for this manuscript is not a dictionary

Line 220 – remove “(Wulf.) Lamouroux” and “(Ag.)” from the scientific names because this is the only place you decided to put them (these are the names of the authors that identified the species referred, and you may or may not write them, but if you do write them once, you have to write them always).

Anticoagulant and antithrombotic activities

Line 234,235, 236 – galactans and carragenans should be plural

Antidiabetic activity

Line 247 – correct and write the full scientific name of G. opuntiav

Line 250 – correct the sentence

Line 254 – remove “”

Anti-obesity activity

Line 268 – some words are in italic

Antihypertensive activity

Line 282 – it seems that some words are missing from the end of the sentence.

Acetylcholinesterase inhibitory ''Alzheimer's Disease''

Line 293 – include (AchE)

Line 298 – write the full scientific name of H. musciformis

Line 302 – The correct name for (2) is P. capillacea. Use full scientific name for (3) O. secundiramea.

Macroalgae for skincare

Line 315 – remove “”; use full scientific names.

Line 317 – be aware that, although the authors claim the use of such species in skincare, if A. armata is Asparagopsis armata, this is a toxic species, and thus its use in cosmetics is doubtful. This is why you must be more critical regarding what you read. Please rephrase.

Lines 322-324 – this sentence makes sense in the introduction, not here. It has nothing to do with skincare.

Conclusions

326 – medical and cosmetic?

329 – are already in use mainly in food industry

Line 332 – correct Jaina to Jania

Author Response

Reviewer 2

 Comments and Suggestions for Authors

General remarks:

A worthy effort on an important area of research, providing consistent information regarding medical and cosmetic applications of Rhodophytes.

The references cited are recent and balanced, although there is so much information regarding the subject that many other references exist and could have been cited.

Some English reviewing is required because some sentences are difficult to understand.

Also, most sections seem just a compilation of sentences, copied from different papers, with no critical analysis from the authors. I believe, thus, that a major effort must be made to review the paper, rephrasing the paragraphs, and making a more mature/critical judgment of the information searched.

Also, in each section, a few species are mentioned as examples of the bioactivities referred, which is correct. But the way the text is written, it seems that only those species have such bioactivities. I suggest some rephrasing in each section, addressing this issue.

Thus, although I acknowledge the effort and the importance of the matter, I recommend a major revision of the paper.

Specific remarks are made below to improve the manuscript.

 Specific remarks.

Abstract

Remove the “” from Rhodophyta and from carrageenans and agars

Done

Introduction

Line 33. I disagree that red seaweed are the “best critical source” for the other two groups also present very important compounds.

Indeed, green and brown algae contain many active substances, but in the medical field red algae represent the best group due to their high polysaccharides content, phycobiliproteins content which exhibited many biological activities.

I suggest you remove the word “best”.

Done

Line 40. Your examples are rather strange. I suggest you rephrase into “animal or plant origin such as meat, eggs or leguminous”.

Done

Line 42. I disagree with the statement “Red and green seaweed contain the largest proportion of phenolic compound”. And that is not what reference [6] says: “The largest proportion of phenolic compounds contained in green and red algae are bromophenols, phenolic acids, and flavonoids.” Actually, brown usually have the highest content in phenolics. I suggest rephrasing.

While, phlorotannins, are the major polyphenolic secondary compounds synthesized only in marine brown seaweed [6].

Line 52 – You should make a general statement about the ability of the red seaweeds to control/kill microbes. The way you wrote it, it seems that only a few species have an antimicrobial activity which is not true – many others also have it and are not mentioned in the text.'

  • Of course, there are many red species that have an antimicrobial ability, which we are not able to mention all, but we focused on the special active substances characterized the red algae. Some references were added.
  • Lines 53-55 – please rephrase into a more correct English form, because I can’t understand what you mean.
  • Done
  • Line 57 – correct Jaina to Jania

    Done
  • Line 58 – Who are “they”?'

    • Done

    Line 65 – Actually new molecules with antimicrobial and cytotoxic activities were identified by Rodrigues et al 2015 for Sphaerococcus coronopifolius (Mar. Drugs 201513(2), 713-726; https://doi.org/10.3390/md13020713). This is an interesting reference to cite both in this section and in the anticancer section.

    • It is cited two times at line 72 and 188.

    Line 70 – the sentence makes no sense with “33.3%” and “8.3”. What do you mean?

    • This is ratio of antifungal activity of different seaweed groups.

    Line 71 – because it is the first time you cite Cy. crinita you must write the full scientific name (the two epithets).

    -This sentence has been rewritten

     Antiviral activity

    Line 89 – remove “”; both agars and carrageenans should be plurals because there are different molecules of each produced by red seaweeds.

    • Done

    Line 91 – correct obtuse to obtusa

    • Done

    Line 96 – remove “” from the species name Solieria filiformis

    • Done
    • Line 97 – Bromoform, produced by Asparagopsis sp. also has interesting antiviral properties that you may refer (besides reducing methane production).

      • Most of the research focuses on the antiviral activity of polysaccharides from Asparagopsis not bromoform since but it is probable human carcinogen (U.S. 1999) but it paly an essential role in inhibiting methane production under liming concentrations.
      • U.S. Environmental Protection Agency. Integrated Risk Information System (IRIS) on Bromoform. National Center for Environmental Assessment, Office of Research and Development, Washington, D.C. 1999.

      Antioxidant activity

    • Line 105-109 – remove all “”

    • Done
    •  

      • Line 110 – again, Carrageenan, porphyrin, and agar should be plurals
      • Done

       Line 116-117 – this sentence doesn’t add any new information from the sentence in line 110. I suggest you remove it.

      -Done.

      Line 118 – What do you mean by “one of the best groups of pigment proteins from red macroalgae”? As far as I know, red seaweeds present only this group of protein pigments. I suggest rephrasing.

      -Done

      Line 124 – If you decide to keep the sentence in lines 116-117 “Reactive Oxygen Species“ should go up there.

      -Done

      Line 126 – DPPH, as ABTS, TPC, or FRAP, are methods to assess antioxidant ability, so antioxidant compounds are measured using such methods. I don’t really understand what you mean by this sentence.

      • Sure, these are methods for estimation the antioxidant activity of each compound. This sentence means the positive relationship between phenolic and flavonoid contents and free radical scavenging activity of algal extract which confirmed by DPPH assay.

      Line 132 – remove “” and substitute by (). Rephrase the sentence into a more correct English.

      -Done

      Line 135 – substitute P. capillacea by full scientific name.

      • Done

      Line 136 – 50% of what??

        - 50% AsA (Ascorbic Acid)

      Line 139 – correct Parmaria palmate to Palmaria palmata

      • Done

      Anticancer and antiproliferative activities

      Line 146 - – remove “” from both species names

      • Done

      Line 163 – the sentence “They are also used as fluorescent indicators in immunological and clinical diagnostics” makes no sense in the context of this section. I suggest deleting it.

      • Done

       Line 187 – the sentence “Mycosporine from red seaweed had antioxidant activities [3]” makes sense in the antioxidant section, not here.

      • Done

       Anti-inflammatory activity

      Line 193 – link references 53 and 56, because the anti-inflammatory effect of G. verrucosa (that currently is known as Gracilariopsis longissima) may be due to the presence of R-phycoerythrin.

      - Gunerken [53]  detected the anti-inflammatory and antioxidant capacities of Gracilaria verrucosa related to polysaccharide fractions not R-phycoerythrin.  

      Analgesic and antipyretic activities

      Line 209 – rephrase the first sentence for this manuscript is not a dictionary

      • Done

       Line 220 – remove “(Wulf.) Lamouroux” and “(Ag.)” from the scientific names because this is the only place you decided to put them (these are the names of the authors that identified the species referred, and you may or may not write them, but if you do write them once, you have to write them always).

      • Done

        Anticoagulant and antithrombotic activities

      Line 234,235, 236 – galactans and carragenans should be plural

      • Done

      Antidiabetic activity

      Line 247 – correct and write the full scientific name of G. opuntiav

      • Done

      Line 250 – correct the sentence

      • Done

      Line 254 – remove “”

      • Done

       Anti-obesity activity

      Line 268 – some words are in italic

      - in vivo usually type in italics form.

       Antihypertensive activity

      • Line 282 – it seems that some words are missing from the end of the sentence.
      • Done

      Acetylcholinesterase inhibitory ''Alzheimer's Disease''

      Line 293 – include (AchE)

      • Done

      Line 298 – write the full scientific name of H. musciformis

      • Done

      Line 302 – The correct name for (2) is P. capillacea. Use full scientific name for (3) O. secundiramea.

      • Done

       Macroalgae for skincare

      Line 315 – remove “”; use full scientific names.

      • Done

      Line 317 – be aware that, although the authors claim the use of such species in skincare, if A. armata is Asparagopsis armata, this is a toxic species, and thus its use in cosmetics is doubtful. This is why you must be more critical regarding what you read. Please rephrase.

      • Indeed, there are products that are made from Asparagopsis armata extract "AldvineTm 5x and Phykosil for anti-aging and anti-wrinkle" in the global markets (https://cosmetics.specialchem.com/inci/asparagopsis-armata-extract). From our point of view, the algal extract is treated and purified for obtaining safe product.
      • A. armata is not toxic specie; it is an invasive species which competitive advantage arises from the production and release of large amounts of toxic compounds to the surrounding invaded area, reducing the abundance of native species for protection itself (Silva et al., 2020).

      Moreover, Asparagopsis armat is used as a feed additive to reduce enteric methane emissions (Roque et al., 2019).

      Silva, C.O.; Novais, S.C.; Soares, A.M.; Barata, C.; Lemos, M.F. Impacts of The Invasive Seaweed Asparagopsis armata Exudate on Rockpool Invertebrates. Preprints 2020

      Roque, BM., Salwen, J.K., Kinley R., Kebreab E. (2019). Inclusion of Asparagopsis armata in lactating dairy cows’ diet reduces enteric methane emission by over 50 percent. Journal of Cleaner Production.234: 132-138.

      • Lines 322-324 – this sentence makes sense in the introduction, not here. It has nothing to do with skincare
      • Removed 

       Conclusions

      326 – medical and cosmetic?

      • Done

      329 – are already in use mainly in food industry

      • Done

      Line 332 – correct Jaina to Jania

      • Done

Round 2

Reviewer 2 Report

The review article was fully reviewed, with most of the suggestions made.

All unclear issues were scaled up and corrected. The text is therefore more consistent and clear. It presents information relevant to the scientific community and describes the medical and cosmetic properties of red seaweeds in a very comprehensive way.

With the corrections made, I believe it is in a position to be published with 4 minor changes.

Line 139: correct name to Pterocladia

Line 143: correct name to palmata

Line 271 – some words are in italic

Line 318 - correct name to Gelidium